# Baseline Correction of Acceleration Data Based on a Hybrid EMD–DNN Method

**DOI:** 10.3390/s21186283

**Published:** 2021-09-19

**Authors:** Zengshun Chen, Jun Fu, Yanjian Peng, Tuanhai Chen, LiKai Zhang, Chenfeng Yuan

**Affiliations:** 1Institute of Engineering Mechanics, China Earthquake Administration, Harbin 061019, China; zchenba@connect.ust.hk; 2School of Civil Engineering, Chongqing University, Chongqing 400045, China; 201916021048@cqu.edu.cn (J.F.); chenfengyuan@cqu.edu.cn (C.Y.); 3Research and Development Center, Gas & Power Group, CNOOC, Beijing 100028, China; pengyj4@cnooc.com.cn (Y.P.); chenth4@cnooc.com.cn (T.C.)

**Keywords:** empirical mode decomposition, deep neural network, baseline drift, baseline correction, displacement measurement

## Abstract

Measuring displacement response is essential in the field of structural health monitoring and seismic engineering. Numerical integration of the acceleration signal is a common measurement method of displacement data. However, due to the circumstances of ground tilt, low-frequency noise caused by instruments, hysteresis of the transducer, etc., it would generate a baseline drift phenomenon in acceleration integration, failing to obtain an actual displacement response. The improved traditional baseline correction methods still have some problems, such as high baseline correction error, poor adaptability, and narrow application scope. This paper proposes a deep neural network model based on empirical mode decomposition (EMD–DNN) to solve baseline correction by removing the drifting trend. The feature of multiple time sequences that EMD obtains is extracted via DNN, achieving the real displacement time history of prediction. In order to verify the effectiveness of the proposed method, two natural waves (EL centro wave, Taft wave) and one Artificial wave are selected to test in a shaking table test. Comparing the traditional methods such as the least squares method, EMD, and DNN method, EMD–DNN has the best baseline correction effect in terms of the evaluation indexes: Mean Absolute Error (MAE), Mean Square Error (MSE), Root Mean Square Error (RMSE), and degree of fit (R-Square).

## 1. Introduction

For civil structures, the damage of a structure under earthquake or strong wind can be judged and the seismic or wind resistance of the structure can be evaluated by displacement responses [1,2]. Displacement is not only an important design control parameter, but also an important working condition index; therefore, displacement measurement is an indispensable part of structural seismic design in the field of earthquake engineering, and also an important part of structural health monitoring. In the field of earthquake engineering, on the one hand, the shaking table laboratory can obtain the real seismic response of the structure. The dynamic testing instruments widely used in shaking table tests are acceleration sensors and pull-wire displacement sensors. Due to the limitation of the number of channels in the acquisition system, it is complicated to collect a large amount of acceleration and displacement data simultaneously. Therefore, numerical integration of acceleration data is the main method to obtain structural displacement response. On the other hand, strong vibration acceleration records provide valuable basic information for seismologists and engineers to study the mechanism of focal rupture in large earthquakes and the failure mechanism of engineering structures, in which displacement data need to be obtained by numerical integration of acceleration records. In practical structural health monitoring, searching for a relatively fixed reference point for most outdoor structures is hardly realized because of the fixed displacement sensors. Therefore, many displacement measurement techniques are not suitable for structural displacement monitoring in practice. Although the Global Positioning System (GPS) does not require fixed reference points, monitoring points can only be arranged at essential locations where the displacement range is much greater than the accuracy of the GPS. The cost is expensive and the accuracy is relatively low [3]. Compared with displacement monitoring technology, acceleration monitoring technology is mature, accurate, low-cost, and does not require a fixed reference point. Therefore, for seismic engineering and structural health monitoring, displacement data need to be obtained through acceleration data integration. However, due to the following reasons, the direct numerical integration of acceleration records will cause baseline drift. The reasons for baseline drift are summarized as follows:(1)The ground rotates or tilts [4];(2)The recorded data is mixed with low-frequency noise from environmental vibration or the vibration of the instrument itself;(3)The assumed initial value of velocity or displacement is inconsistent with the actual situation;(4)Data processing errors and transducer hysteresis [5].

In order to eliminate the baseline drift and obtain the real displacement response of the structure, various baseline correction methods have been proposed. At present, baseline correction methods can be divided into three categories:(1)Eliminate displacement low-frequency noise error by filtering method [6,7,8];(2)The displacement is corrected piecewise based on the corrected acceleration and velocity [5,9,10,11];(3)Eliminate drift error by eliminating velocity and displacement polynomial trend.

Strong earthquake records play an important role in seismic engineering design, but are often affected by low-frequency noise, limiting the frequency ranging from useful data. In strong motion records, compared with intermediate frequencies, the influence of noise is most obvious at low frequencies, in which the signal-to-noise ratio is low [12]. Therefore, Chiu proposed that the application of a high-pass filter can effectively eliminate baseline errors related to low-frequency components [6]. However, the high-pass filter does not only eliminate baseline errors but also low-frequency signal content, including permanent ground displacement, and reduces dynamic ground displacement [10].

The piecewise correction method based on corrected acceleration and velocity was first proposed by Iwan et al. [5], who assumed that the baseline drift of strong earthquake records was caused by the tiny mechanical vibration or electrical lag that occurred in the most intense part of the ground motion in the transducer system. The basic idea of Iwan’s method was to divide the whole acceleration time history into three stages: before earthquake, strong earthquake, and postearthquake. Two constant acceleration values were used to modify the acceleration records of the strong earthquake and postearthquake segments, and the final displacement was obtained by quadratic numerical integration of the corrected acceleration records. Since then, there have been many improved methods for Iwan, such as the V0 correction method proposed by Boore et al. [9]; Wu and Wu’s method [11]; and an automatic iteration method proposed by Wang et al., which was widely recognized in earlier years [10]. Although all these methods have been applied, they have advantages and disadvantages in the process of processing, which is mainly reflected in the following aspects:(1)Some methods rely on subjective experience judgment, and the uncertainty of correction is large;(2)The process of some methods is complicated, and the calculation efficiency is low. The summary comparison of these methods is shown in Table 1.

The polynomial detrending algorithm is the most widely used baseline correction method in recent years. This algorithm uses the least squares method to fit the displacement time-history curve obtained by numerical integration of acceleration, thereby removing the polynomial trend from displacement data. Antoniou [13] introduced this algorithm into the ground motion signal processing software “Seismosignal” in 2015 to adjust the baseline of strong earthquake data. However, the traditional detrending algorithm needs several trials to determine the most appropriate order of the fitting polynomial, and relatively high-order fitting polynomials may overcorrect the signal, resulting in an inability to predict and control the profile of the corrected velocity and displacement.

In order to control the baseline correction results directly, Chao Pan [14] proposed a target-based baseline correction algorithm that directly equalizes or approximates the acceleration, velocity, and displacement values with the predetermined target values for the acceleration signals whose initial velocity and displacement are inconsistent. However, this method was adjusted several times to adapt the number of basis functions M [14]. Moreover, the acceleration curve with a very large baseline drift cannot be corrected.

As the traditional baseline correction method will cause the corrected final displacement to be extremely large, Lin [15] proposed a method of baseline correction for near-fault strong earthquake records based on the target final displacement. Taking the offset of the traditional baseline correction displacement and the final displacement of the target as the index, the acceleration time history of the ground motion record is adjusted (correcting the pseudostatic component of the ground motion record), and the final displacement of the target is obtained by quadratic integration so that the final displacement is consistent with the final displacement of the target. However, this method is not adaptive, nor can it eliminate or minimize the errors caused by baseline drift.

Most of the current studies have improved the traditional baseline correction methods to varying degrees. Still, they all have some problems, such as high baseline correction error, poor adaptability, and narrow application scope. In order to improve the accuracy of baseline correction and establish a widely applicable method to eliminate baseline drift, this paper proposes a deep neural network model based on empirical mode decomposition (EMD–DNN) to correct the baseline of acceleration records.

## 2. Traditional Baseline Correction Method

The least squares method (LSM) is the most widely applicable method for baseline correction [16]. The acceleration, velocity, and displacement of a certain signal are, respectively, expressed as a(t), v(t), and l(t), and the relationship between them can be expressed as
(1)$$\begin{array}{c} v\left(t\right)=v_{0}+\int_{t_{0}}^{t}a\left(t\right)dt \end{array}$$
(2)$$\begin{array}{c} l\left(t\right)=l_{0}+\int_{t_{0}}^{t}v\left(t\right)dt \end{array}$$
(3)$$\begin{array}{c} =l_{0}+v_{0}\left(t-t_{0}\right)+\int_{t_{0}}^{t}\int_{t_{0}}^{t}a\left(\tau \right)d\tau dt \end{array}$$
where t is the time variable; and $t_{0}$, $a_{0}$, $v_{0}$, and $l_{0}$, respectively, represent the initial values of time, acceleration, velocity, and displacement. Since the actual initial velocity $v_{0}$ and initial displacement $l_{0}$ are difficult to obtain, the assumed initial velocity $v_{0}$ and initial displacement $l_{0}$ are adopted in the integration of acceleration, so as to obtain the time-history data of velocity and displacement. The pseudovelocity and pseudodisplacement obtained from the assumed initial value can be written as
(4)$$\begin{array}{c} v^{\prime }\left(t\right)=v_{0}^{\prime }+\int_{t_{0}}^{t}a\left(t\right)dt \end{array}$$
(5)$$\begin{array}{c} l^{\prime }\left(t\right)=l_{0}^{\prime }+\int_{t_{0}}^{t}v^{\prime }\left(t\right)dt \end{array}$$
(6)$$\begin{array}{c} =l_{0}^{\prime }+v_{0}^{\prime }\left(t-t_{0}\right)+\int_{t_{0}}^{t}\int_{t_{0}}^{t}a\left(\tau \right)d\tau dt \end{array}$$

Then, the error between the velocity and displacement results and the actual results is obtained according to Equations (1)–(6).
(7)$$v_{error}(t)=v^{\prime }(t)-v(t)=v_{0}^{\prime }-v_{0}$$
(8)$$l_{error}(t)=l^{\prime }(t)-l(t)$$
(9)$$=\left(d_{0}^{\prime }-d_{0}\right)+\left(v_{0}^{\prime }-v_{0}\right)\left(t-t_{0}\right)$$

Through the shaking table test in Chongqing University, we compressed seismic waves in time history according to the scale theory (this will be explained in detail later); then, we loaded the model with a 0.5-g peak acceleration EL centro wave in the X direction and obtained the seismic acceleration response (Figure 1a). It was found that the numerical integration of the obtained acceleration would lead to constant errors in velocity data (Figure 1b) and linear errors in displacement data (Figure 1c). This is consistent with Equations (5) and (6).

The specific process of least squares baseline correction is as follows:(1)The quadratic value of acceleration is $l^{\prime }(t)$;(2)The least squares method is used to fit the linear error trend $l_{error}(t)$ of $l^{\prime }(t)$;(3)The final actual displacement is obtained.
(10)$$l(t)=l^{\prime }(t)-l_{error}(t)$$

## 3. Baseline Correction Method Based on EMD–DNN

### 3.1. EMD

Empirical Mode Decomposition (EMD) is an analysis method for nonstationary signals proposed by Huang [17]. The EMD decomposes the signal according to the time scale characteristics of the data itself, without presetting any basis function or predetermining the frequency range of the trend term, which is essentially different from Fourier decomposition and wavelet decomposition based on an a priori harmonic basis function and wavelet basis function. Due to this characteristic, EMD can be applied to any type of signal decomposition in theory, so it has a significant advantage in processing nonstationary and nonlinear data. Therefore, EMD was quickly and effectively applied in different engineering fields as soon as it was proposed.

In essence, EMD stabilizes no-stationary signals, decomposes the fluctuations and trends of different scales in the signals step by step, and generates a series of data sequences with different characteristic scales. Each sequence is called an inherent mode function IMF, and each IMF should meet the following conditions [18]:(1)In the whole time series, the number of extreme points is equal to the number of zero crossings or the difference is, at most, 1.(2)At any point, the average of upper and lower envelopes is 0. The EMD process is implemented through a process called “screening”. Its specific treatment methods are as follows:

Given the original signal x(t), find all maximum points of x(t) and use the cubic spline function to fit the upper envelope of the original signal. Find all the minimum points of x(t) and use cubic spline function to fit the lower envelope of the original signal; the mean of the upper and lower envelope is the mean envelope of the original signal $m_{1}(t)$; after subtracting $m_{1}(t)$ from the original signal x(t), a new signal $d_{1,1}(t)$ can be obtained to judge whether $d_{1,1}(t)$ meets the two conditions of IMF. If not, $d_{1,1}(t)$ will continue to “screening” according to the above method until $d_{1,k}(t)$—the signal after k times of “screening”—satisfies the IMF condition, denoted as the first IMF component of the original signal IMF1. The remaining component $r_{1}(t)$ is obtained by subtracting from the original signal, and the above “screening” $r_{1}(t)$ is continued. After n times of “screening”, the residual signal $r_{n}(t)$ is obtained. The decomposition would stop when $r_{n}(t)$ is a monotonic function. $r_{n}(t)$ is the trend term of the original signal. The original signal can be expressed as
(11)$$x(t)=\sum_{i=1}^{n}\mathrm{IMF}(i)+r_{n}(t)$$

It can be seen from the “screening” process of EMD that, compared with Fourier transform and wavelet decomposition, EMD does not need to set the basis function and has self-adaptability, so it is applicable to a wider range. After the decomposition of the original signal x(t), the first IMF component contains the component with the smallest time scale (the highest frequency) in the original signal x(t). With the increase in the IMF order, its corresponding frequency component decreases gradually, and the frequency component of the remaining residual quantity $r_{n}(t)$ is the lowest. According to the convergence conditions of EMD decomposition, the residual decomposition quantity $r_{n}(t)$ is a monotonic function whose period time will be larger than the recording length of the signal. Therefore, $r_{n}(t)$ can be regarded as the trend term of the original signal x(t).

The specific process of baseline correction through EMD is as follows:(1)The pseudo velocity $v^{\prime }(t)$ can be obtained by the acceleration record integration.
(12)$$v^{\prime }(t)=\int_{t_{0}}^{t}a(t)dt$$(2)Each IMF and trend term $r_{1,n}(t)$ can be obtained through the empirical mode decomposition of pseudo velocity $v^{\prime }(t)$. The correction velocity v(t) is obtained by removing the trend term $r_{1,n}(t)$ and reconstructing the IMF component.(3)The pseudo displacement $l^{\prime }(t)$ is obtained by velocity v(t) integration.
(13)$$l^{\prime }(t)=\int_{t_{0}}^{t}v(t)dt$$(4)The pseudo displacement $l^{\prime }(t)$ is decomposed by the EMD to obtain each and the trend term $r_{2,n}(t)$ (Figure 2). The trending term $r_{2,n}(t)$ is removed, and the correction displacement l(t) is obtained by reconstructing the IMF2(i) components. The formula is as follows:(14)l′(t)=∑i=1nIMF2(i)+r(t)2,n
(15)$$l(t)=\sum_{i=1}^{n}\mathrm{IMF}2(i)$$

Taking the EL centro wave of 0.5-g peak acceleration in X direction loaded by the shaking table on the LNG storage tank model as an example, the displacement obtained from EMD (the blue line in Figure 2g) was compared with the actual displacement data (the red line in Figure 2g) recorded by the pull-wire displacement sensor of the shaking table, and it could be seen that this method has a specific baseline correction effect (Figure 2).

### 3.2. DNN

In recent years, with the rapid development of deep learning, it has been applied in various fields of industry and academia, and the development of the deep neural network is the critical technology among them. Deep Neural Network (DNN) is one of the essential technologies of deep learning [19,20], mainly composed of three neural layers: input layer, hidden layer, and output layer.

The input layer, also known as the visible layer, is the only observable architecture of the neural network model. It is used to receive the preprocessed input information, carry out the preliminary fusion extraction of the input information features through the weight, and transfer the extracted feature information to the next layer.

The hidden layer is the fundamental architecture of DNN model, and it is the leading architecture for the DNN model to extract the fusion information features. However, according to the principle of information theory, in computing and processing, data will lead to the loss of information. They inevitably produce noise, which exists in the features extracted from the hidden layer. Therefore, when the hidden layer extracts data in a higher dimension, although the richness and dimensional information is improved, the noise generated will significantly interfere with the subsequent processing. When the hidden layer extracts the data for dimensionality reduction, the feature information will be lost, although the noise generation is reduced. Therefore, we will reasonably choose the processing method according to the situation of the input data.

The output layer is the output architecture of the DNN model, which is the final output of the original input data from the DNN model. Generally, particular algorithm classifiers, such as Softmax, linear regression, logistic regression, etc., are used for similar input layer processing, and one or multiple neurons are selected. The output weight of the last layer is used to extract the feature information of the highest feature layer and extract the final model calculation results.

The layers of the deep neural network (DNN) are fully connected. The calculation formula of the jth neuron in the k layer is as follows:(16)$$y_{j}^{k}=f(\sum_{\left(i=1\right)}^{H_{\left(k-1\right)}}w_{\left(i,j\right)}^{k}x_{i}^{\left(k-1\right)}+b_{j}^{k})$$

In the formula, $H_{k-1}$ is the number of neurons in the layer k−1; $y_{i}^{k-1}$ is the ith neuron in the layer k−1; $w_{i,j}^{k}$ is the weight between ith neuron and the jth; $b_{j}^{k}$ is the bias of the jth neuron in the k layer; f is the nonlinear activation function.

In addition, there are two main operations in the DNN model: forward propagation and back propagation. In a forward propagation network, information starts from the input layer and moves forward layer-by-layer to the output layer until the deviation between the predicted value and the target is outputted to obtain the value of the loss function. The purposed model training is to reduce the value of the loss function until it approaches its minimum value so that the output result of the model is infinitely close to the target. The direction of its gradient descent is the direction in which the loss function decreases. At the same time, the direct derivative can only obtain the gradient of the loss function relative to the weight of the output layer. It is necessary to use the chain derivative rule to find out the gradient of the weight of each layer step-by-step until the gradient of the loss function relative to the network ownership weight is found. This process is called back propagation [21].

On the basis of back propagation, the weight of the model is constantly updated at a certain learning rate, so that its gradient is constantly declining and the value of the loss function is constantly decreasing, which is called model optimization. Common optimization algorithms include SGD (Stochastic Gradient Descent) [22], Ada Grad (Adaptive gradient algorithm) [23], RMSProp (Root Mean Square Propagation), Adam [24], etc. The training process of DNN model is shown in Figure 3.

### 3.3. EMD–DNN

Due to many factors, such as data processing error, the hysteresis of transducer and mixing of low-frequency noise generated by the vibration of the instrument itself, the vibration response of the structure has the characteristics of fluctuation. Therefore, the response sequence of the structure has nonlinear and nonstationary features to a certain extent, and the improvement of prediction accuracy by using the traditional detrend method is limited. Considering that EMD can obtain more modal characteristics of data and DNN has an excellent performance in time series data modeling, this paper proposes an EMD–DNN prediction model. The process of this method is as follows:(1)The drift displacement time history X(t) was obtained by quadratic numerical integration of the original acceleration time history;(2)Drift displacement time history X(t) was divided into a training set and testing set. The training set and testing set were, respectively, decomposed into multiple IMF components and residual RES by EMD. IMF component and drift displacement time history X(t) of the training set were taken as a feature set to form a combined data set Con1, and the testing set was the combined data set Con2.(3)The combined data set Con1 was input into the DNN model as the input layer. The predicted value was obtained through forward propagation. The loss value was obtained by combining the predicted value with the actual displacement record and acted on the DNN model through back propagation to optimize the model. Finally, the combined data set Con2 was input into the optimal model to evaluate the predicted results. The specific EMD–DNN correction process is shown in Figure 4.

## 4. Experimental Verifications

### 4.1. Experiment Introduction

The test was carried out in the shaking table laboratory of Chongqing University (Figure 5). The length and width of the shaking table were both 6.1 m and the maximum bearing capacity was 60 tons. Taking the 160,000 $m^{3}$ LNG storage tank as the prototype, the length similarity coefficient $S_{l}$ = 1/30, Poisson’s ratio similarity coefficient and strain similarity coefficient were both set as 1, and the acceleration similarity constant $S_{a}$ = 1. The similarity relationship was obtained as follows:(17)$$S_{k}=S_{E}S_{l}$$
(18)$$S_{x}=S_{l}$$
(19)$$S_{a}=S_{x}/S_{t}{}^{2}$$

Among them, $S_{E}$ is the elastic modulus similarity coefficient, $S_{k}$ is the stiffness similarity coefficient, and $S_{x}$ is the displacement similarity coefficient. The acceleration similarity coefficient can be obtained as
(20)$$S_{t}=\sqrt{S_{l}/S_{a}}=\sqrt{1/30}=0.183$$

Therefore, the seismic wave time should be reduced to 0.183 times the original record when the ground motion is input.

#### 4.1.1. Sensor Arrangement

The test equipment and instruments used in the experiment include the 128-channel NI high-speed data acquisition system from the United States, 128-channel DEWETRON high-speed data acquisition system from Austria (the sampling frequency of the sampling channel is 256 HZ), MEAS acceleration sensor from the United States, Endevco acceleration sensor from the United States, Unimeasure cable displacement sensor from the United States, and resistive strain gauge.

(1)Acceleration sensor layout
1.Pile: one acceleration sensor was arranged in each X, Y, and Z direction of the pile head of the isolation tank and the nonisolation tank.2.Inner tank: there were four measuring points along the height direction, respectively, in the isolation tank and the nonisolation tank, and one acceleration sensor was arranged at each measuring point in X, Y, and Z directions, respectively.3.The outer tank: there were seven measuring points on the tank body and its dome (there were two measuring points on the dome and five measuring points on the tank body along the height direction). One acceleration sensor was arranged on each measuring point in the X, Y, and Z directions, respectively.
(2)Displacement meter layoutX directional and Y directional Unimeasure pull-wire displacement meters are arranged on the top of the LNG storage tank dome (Figure 6 and Table 2).

#### 4.1.2. Select Wave Type and Parameter Settings

The peak accelerations of the test loading conditions were taken as 7 degrees of fortification intensity (PGA = 0.1 g), 7.5 degrees of rare encounter intensity (PGA = 0.25 g), 8.5 degrees of rare encounter intensity (PGA = 0.5 g), and 9.5 degrees of rare encounter intensity (PGA = 0.75 g). Wenchuan Wolong wave, EL centro wave, Taft wave, and Artificial wave were used to simulate the seismic test of the model structure. EL centro wave and Taft wave were obtained in the Peer Ground Motion Database according to the designed response spectrum. The Artificial wave was generated by SIMQKE_GR software utilizing the designed reaction spectrum, and the Wenchuan Wolong wave was provided by the shaking table laboratory of Chongqing University. All seismic waves were baseline-corrected and filtered before loading.

Before and after the input of different intensity seismic waves, the model was swept by white noise to test the dynamic characteristics of the model structure, such as natural vibration frequency, mode shape, and damping ratio. According to the similarity relation, the duration of seismic wave was compressed to 0.183 of the original seismic wave. As the storage tank is a symmetrical structure, with the input three directions of ground motion, the Artificial wave peak acceleration ratio in X direction, Y direction, and Z direction was 1:0.85:0.65. When the two directions’ ground motion was inputted, the ratio of the Artificial wave in X direction and Z direction was 1:0.85.

The DNN model adopted four fully connected networks—the neurons of each layer were 125, 128, 64, and 1, respectively—and adopted the ReLU activation function. The initial learning rate was 0.01, and the batch size was set to 256. To reduce the overfitting phenomenon, the method of dropout with a ratio of 0.5 was adopted.

#### 4.1.3. The Evaluation Index

In this paper, the mean square error (MSE), root mean square error (RMSE), mean absolute error (MAE), and fitting degree (R-square) were selected to evaluate the baseline correction accuracy of each model.

(1)Mean absolute error (MAE) [25]
(21)$$\begin{array}{c} MAE\left(y,\hat{y}\right)=\frac{1}{n}\underset{i=1}{\sum^{n}}\;\left|y_{i}-{\hat{y}}_{i}\right| \end{array}$$

This indicator is the mean value of the sum of absolute errors of the corresponding points between the corrected data and the actual displacement data. The decreasing MAE value would enhance the accuracy of the baseline correction model.

(2)Mean square error (MSE) [26]
(22)$$\begin{array}{c} MSE\left(y,\hat{y}\right)=\frac{1}{n}\underset{i=1}{\sum^{n}}\;{∥y_{i}-{\hat{y}}_{i}∥}_{2}^{2} \end{array}$$

The mean square error is a measure that reflects the degree of difference between the actual displacement data and the corrected data; in other words, the expected value of the square of the difference between the parameter estimate and the parameter truth. MSE can evaluate the degree of change of data. The smaller values of MSE represent the better accuracy of the prediction model handling the experimental data.

(3)Root mean square error (RMSE) [27]
(23)$$\begin{array}{c} RMSE\left(y,\hat{y}\right)=\sqrt{\frac{1}{n}\underset{i=1}{\sum^{n}}\;{∥y_{i}-{\hat{y}}_{i}∥}_{2}^{2}} \end{array}$$

Root mean square error, also known as standard error, is the arithmetic square root of the mean square error. In other words, it is the square root of the ratio of the square of the deviation between the actual displacement data and the corrected data, which, in actual measurements, is always finite and the truth value can only be replaced by the most reliable (best) value. RMSE is very sensitive to very large or very small errors in a set of measurements, so the standard error can reflect measurement precision well. This is the reason why RMSE is widely used in engineering measurement.

(4)R-square [28]
(24)$$R-square=1-\frac{\sum_{i=1}^{n}{\left(y_{i}-{\hat{y}}_{i}\right)}^{2}}{\sum_{i=1}^{n}{\left(y_{i}-{\bar{y}}_{i}\right)}^{2}}$$
where $y_{i}$ is the actual displacement data, ${\bar{y}}_{i}$ is the average values of the actual displacement data, and ${\hat{y}}_{i}$ is the corrected data.

The denominator is the sum of the residual square and the numerator is the variance. R-square is a quantity to measure the ability of model fitting degree. The value of R-square ranges from negative infinity to 100%. The value of R-square is close to 100%, which represents the excellent fitting ability.

### 4.2. Results and Analyses

In this experiment, two sets of shaking table acceleration records (EL centro wave and Taft wave) and a set of Artificially simulated acceleration time-history curves (Artificial wave) were selected. In order to verify the applicability of EMD–DNN baseline correction method, three experimental conditions of different loading directions were selected. They are as follows: (1) X direction loading EL centro wave, with PGA 0.75 g as the training sample and PGA 0.5 g as the test sample; (2) X, Z two-direction loading of Taft wave, PGA of 0.75 g as the training sample, PGA of 0.5 g as the test sample; (3) X, Y, Z three directions loading Artificial wave, PGA of 0.1 g as the training sample, PGA of 0.25 g as the test sample.

Taking the Taft wave with a peak acceleration of 0.5 g in the X, Z directions as an example, the measured acceleration time history and amplitude spectrum are shown in Figure 7.

EMD decomposed three test waves and three training waves, and the decomposition process is as follows (Figure 8, Figure 9 and Figure 10):

As can be seen in Figure 8, Figure 9 and Figure 10, different seismic waves are decomposed by EMD into multiple IMF components and RES residual terms, and each IMF component has a different model. Moreover, the IMF modal component obtained by the EMD is arranged in the order of decreasing frequency, and the number of extreme points of the two IMF components before and after the EMD is almost double the relationship; simultaneously, the frequency band size is decreasing. By removing the RES trend term and reconstructing the IMF component, the low-frequency noise error caused by environmental vibration or the vibration of the instrument itself can be eliminated.

The different IMF components obtained from the EMD are the different modal characteristics of the displacement data. The various IMF components and the displacement time-history curves of the drift are taken as the training sample set and input to the DNN model for training and prediction.

For each dataset, four baseline correction models were established: least squares method, EMD, DNN, and EMD–DNN. The fitting curves are shown in Figure 11, from top to bottom: El Centro wave, artificial wave, and Taft wave.

It can be seen from Figure 11 that the displacement time-history curves fitted by different methods have a certain baseline correction effect. However, it can be found from Figure 11 that the displacement curve fitted by the least squares method and the EMD has a larger dispersion and has a larger offset value compared with the real displacement. Figure 11 shows that the DNN detrend method will weaken the characteristics of the curve, resulting in a smaller value at the peak compared with the real value. In the three figures, the displacement curves obtained by EMD–DNN are the closest to the real displacement in the time history and have the highest degree of fitting.

The evaluation index diagrams of different methods are as follows (Figure 12):

As can be seen from Figure 12, the mean absolute errors value of the least squares method in Artificial wave and EL centro wave are small, but the MAE value in Taft wave is larger than that of the EMD–DNN, which is 2.0145 larger than that of EMD–DNN. This verifies the idea of Yuanzheng Lin [15] that the least squares method can only solve the baseline drift caused because the assumed initial value of velocity or displacement is not consistent with the actual situation. The MAE value of the traditional EMD is higher in the three waves, and the MAE value of the DNN is lower than that of the EMD. In general, the mean absolute error of the EMD–DNN is the smallest compared with other correction methods, indicating that the EMD–DNN is improved to different degrees on the basis of the traditional EMD and the DNN only. In EL centro wave, EMD–DNN was 0.3016 smaller than DNN; in the Artificial wave, EMD–DNN is 0.0676 smaller than DNN. In the Taft wave, EMD–DNN is 0.4614 smaller than DNN. Compared with other correction methods, the MAE of EMD–DNN in EL centro wave is the smallest, and the value is 0.8625.

It can be seen from Figure 13, similar to the mean absolute error (MAE), the MSE value of the least squares method in Artificial wave and EL centro wave is small, but the value of the least squares method in Taft wave is larger. In general, the MSE of EMD–DNN is the smallest compared with other correction methods. In EL centro wave, EMD–DNN is 0.6578 less than DNN. In Artificial wave, EMD–DNN is 0.1278 smaller than DNN. In the Taft wave, EMD–DNN is 6.3582 smaller than DNN. Compared with other correction methods, the mean squared error (MSE) of EMD–DNN in the EL centro wave is the smallest at 1.3648.

As shown in Figure 14, the root mean squared errors of the least squares method in both artificial wave and EL centro wave are small, but the value of the least squares method in Taft wave is larger. In general, the root mean square error of EMD–DNN is the smallest compared with other correction methods. In EL centro wave, EMD–DNN is 0.254 less than DNN. In artificial wave, EMD–DNN is 0.049 smaller than DNN. In the Taft wave, EMD–DNN is 3.8759 smaller than DNN. Compared with other correction methods, MSE of EMD–DNN in the EL centro wave is the smallest, and the value is 1.1682.

As can be seen in Figure 15, for all waves, the fitting degree of least square method is poor, indicating that the correction effect of least square method in the time history of drift displacement is poor, and the fitting degree of EMD is better than that of least square method in the time history of displacement. However, compared with the traditional EMD and least square method, DNN and EMD–DNN greatly improve accuracy, and EMD–DNN has the highest fitting degree. In EL centro wave, EMD–DNN is 20.83% higher than DNN. In Artificial wave, EMD–DNN is 1.86% higher than DNN. In the Taft wave, EMD–DNN is 20% higher than DNN. The highest fitting degree of EMD–DNN in artificial wave is 76.14%.

As can be seen in Figure 16, Figure 17 and Figure 18, for EL centro wave and Taft wave, the traditional least squares method and EMD have high dispersion and poor fitting degree. While the DNN method has a great improvement in fitting effect compared with the least squares method and EMD, the EMD–DNN has the best fitting effect.

As can be seen in Figure 18, for Artificial waves, the fitting effects of the four methods perform well, and EMD–DNN has the best fitting effect compared with other correction methods.

## 5. Concluding Remarks

(1)The traditional least square method has a narrow scope of application and high discreteness. It is mostly used to solve the linear baseline drift caused by the inconsistency between the assumed initial value of velocity or displacement and the actual situation but cannot correct the acceleration error. The EMD applies to a wide range. The structural health monitoring data and shaking table data have specific baseline correction effects using the EMD. Both linear and nonlinear polynomial trends can be eliminated by removing RES items and reconstructing IMF by EMD.(2)The characteristics of drift displacement can be extracted directly by DNN, and the time history of drift displacement is taken as the sample set and input to the DNN model for training and prediction. From the evaluation index, it is shown that this method has a certain prediction effect on the real displacement response.(3)DNN can extract the features of multi-time-series IMF obtained by EMD decomposition, taking displacement time-history curves of different IMF components and drifting as the sample set, these are inputted into the DNN model for training and prediction; finally, improving real displacement response prediction accuracy. It can be obtained from various evaluation indexes that EMD–DNN has the highest accuracy compared with other methods.(4)In practical engineering applications, the EMD–DNN model can be trained only with the data of one displacement monitoring point to predict the actual displacement response of other positions. With this method, the sensor layout can be optimized in the shaking table test and actual structure monitoring, so as to reduce the number of displacement meter layouts and reduce the cost.(5)Since the EMD–DNN model needs to be obtained by real displacement training, we can conduct short-term displacement measurements for a monitoring point of the structure. After the EMD–DNN model is obtained by training, the real displacement response of each position of the structure can be predicted in the long term by the acceleration sensor, without the need of displacement sensor arrangement.

## Figures and Tables

**Figure 1 sensors-21-06283-f001:**
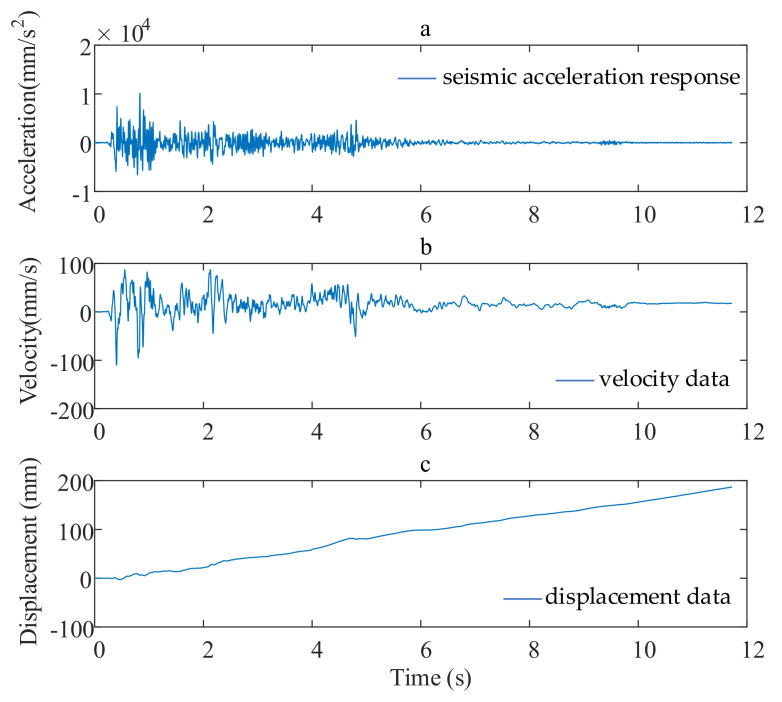
Baseline drift phenomenon of measured acceleration data of shaking table: (**a**) the seismic acceleration response; (**b**) velocity data; (**c**) displacement data.

**Figure 2 sensors-21-06283-f002:**
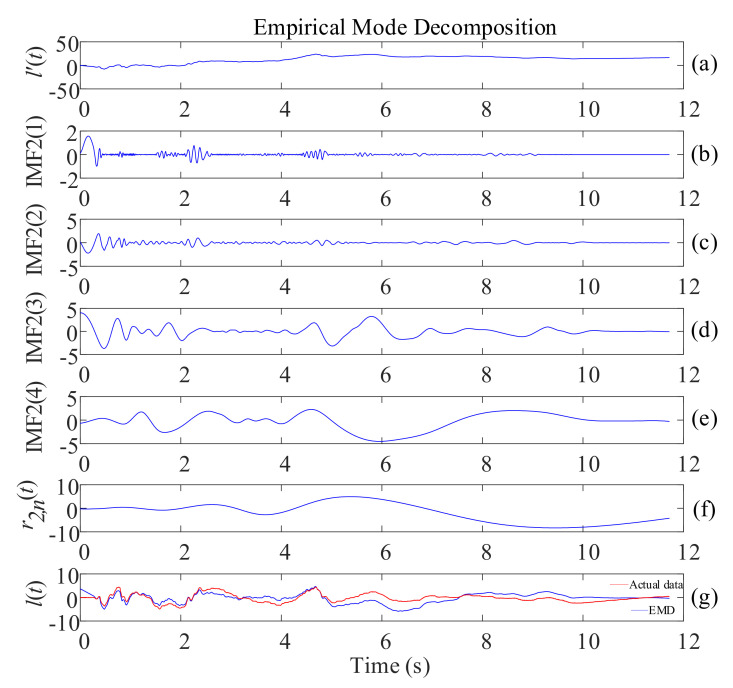
Example of EMD: (**a**) pseudo displacement; (**b**–**e**) different IMFs of the signal; (**f**) residual of the signal; (**g**) the displacement obtained from EMD and the actual displacement data.

**Figure 3 sensors-21-06283-f003:**
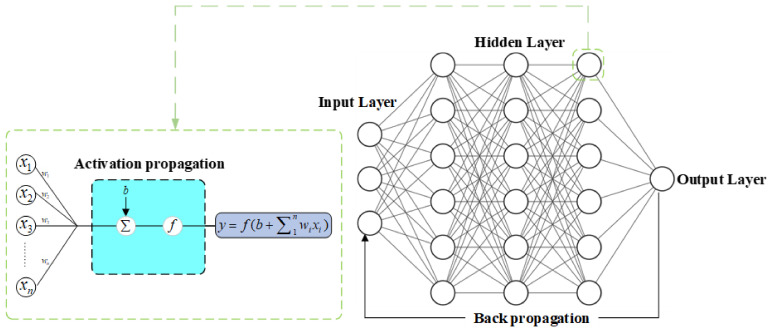
Training flow chart of DNN model.

**Figure 4 sensors-21-06283-f004:**
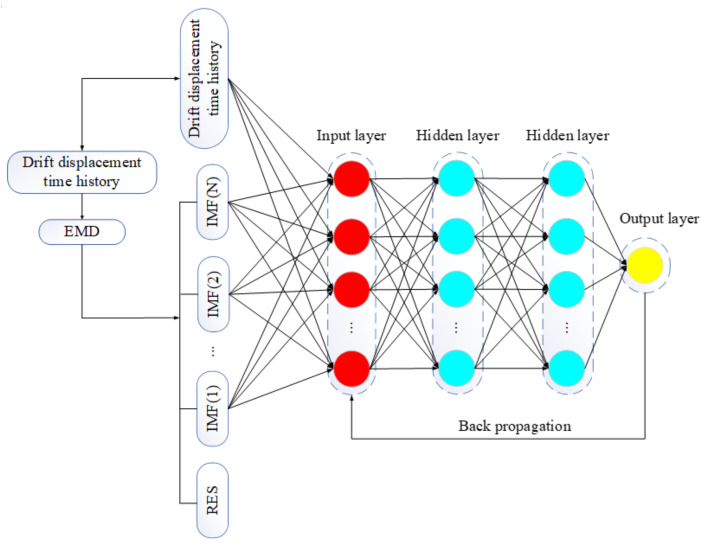
EMD–DNN baseline correction model.

**Figure 5 sensors-21-06283-f005:**
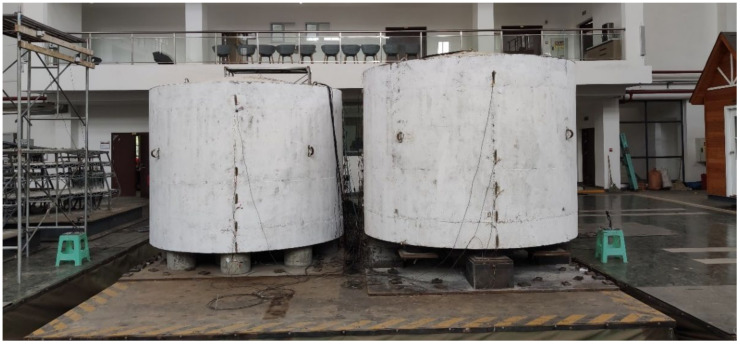
LNG storage tank model.

**Figure 6 sensors-21-06283-f006:**
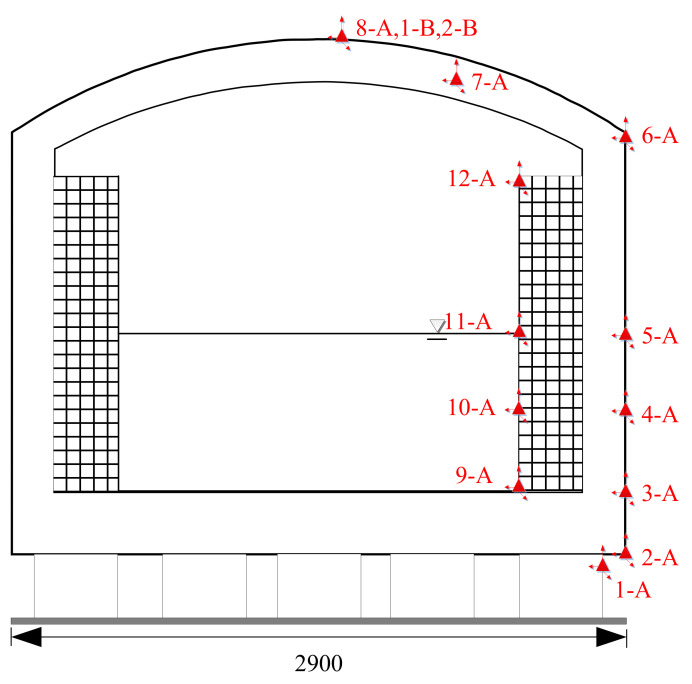
Sensor layout.

**Figure 7 sensors-21-06283-f007:**
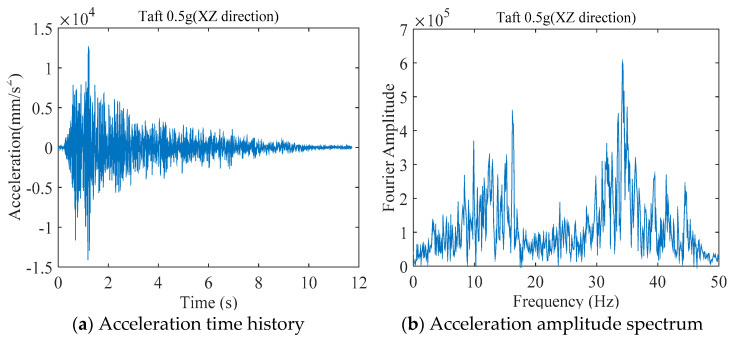
The measured acceleration record of the shaking table.

**Figure 8 sensors-21-06283-f008:**
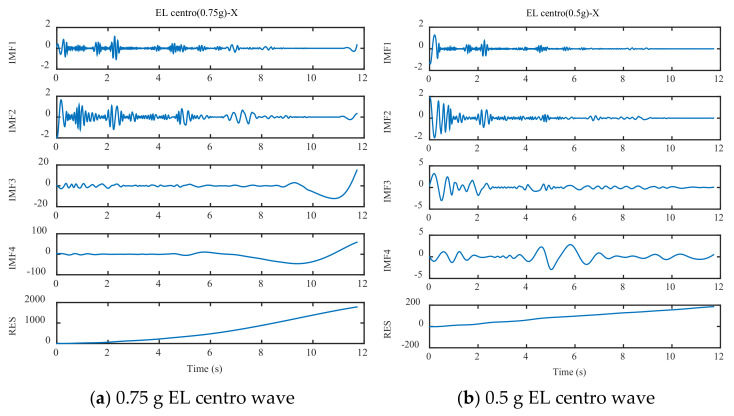
EMD decomposition of EL centro wave.

**Figure 9 sensors-21-06283-f009:**
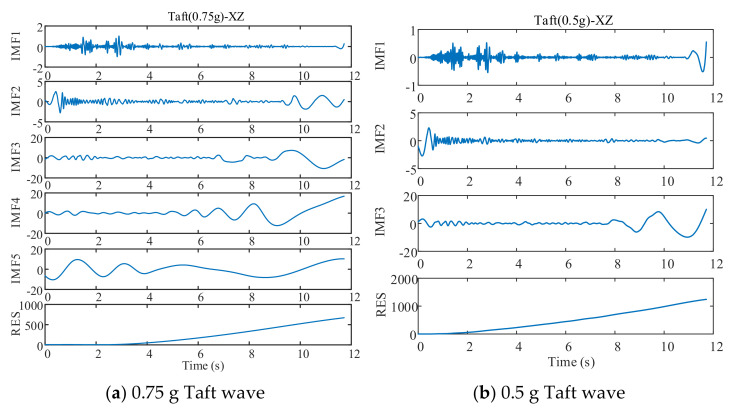
EMD decomposition of Taft wave.

**Figure 10 sensors-21-06283-f010:**
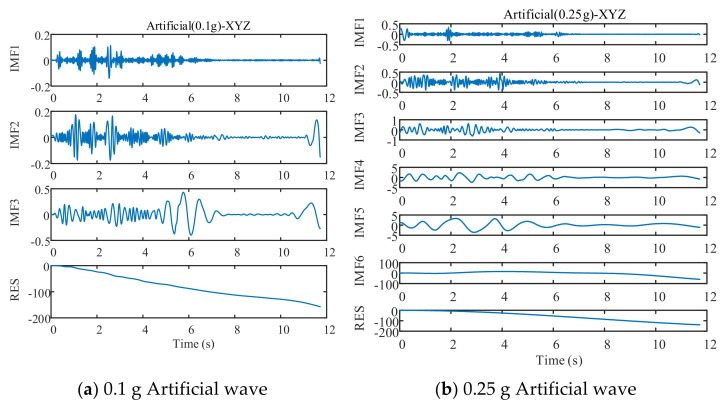
EMD decomposition of Artificial wave.

**Figure 11 sensors-21-06283-f011:**
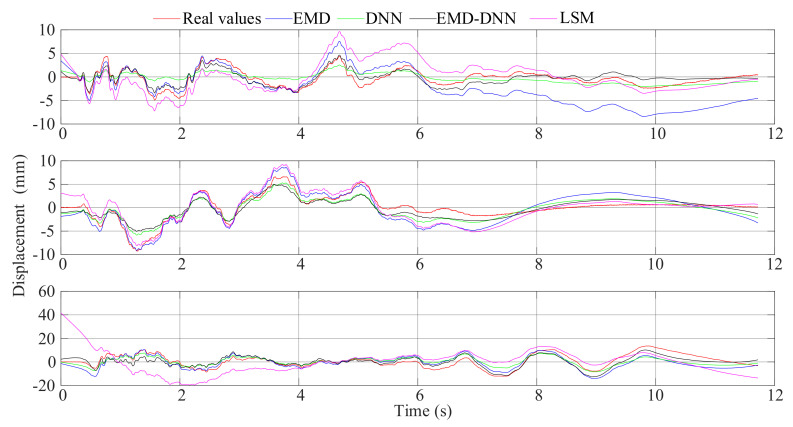
Comparison between baseline correction displacement curves of various methods and real displacement.

**Figure 12 sensors-21-06283-f012:**
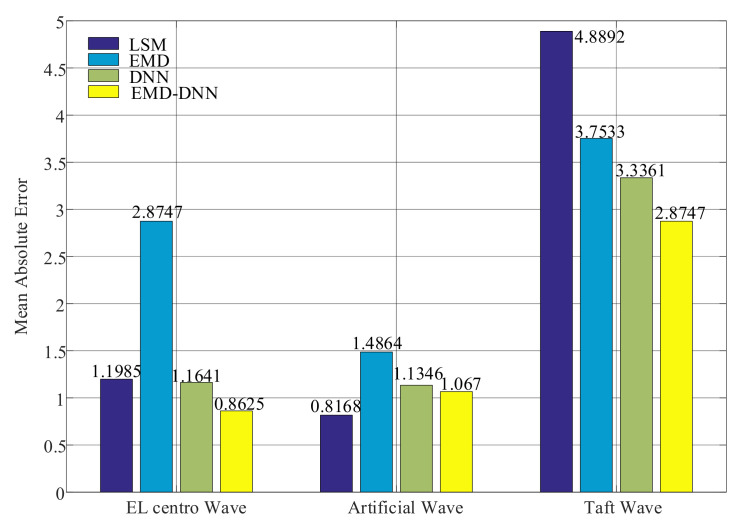
Mean absolute error (MAE) of different correction methods.

**Figure 13 sensors-21-06283-f013:**
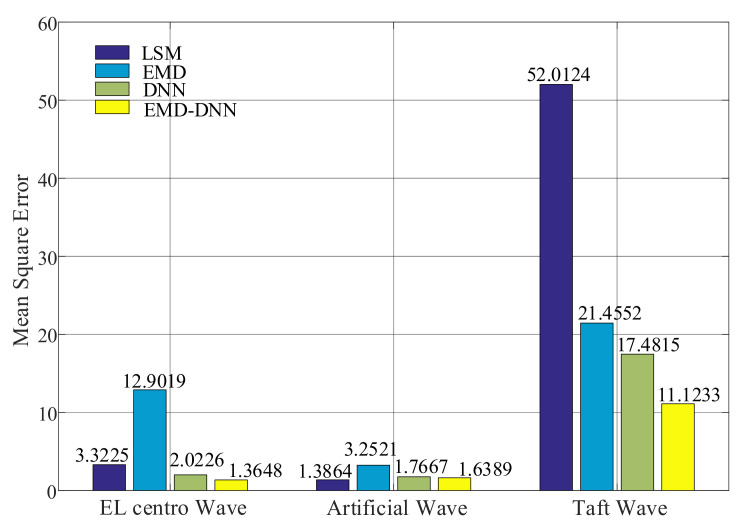
Mean square error (MSE) of different correction methods.

**Figure 14 sensors-21-06283-f014:**
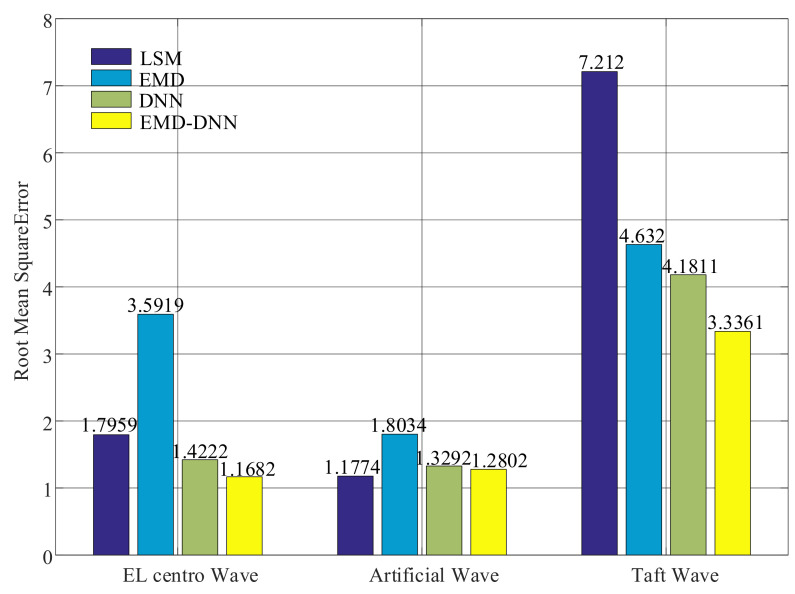
Root mean squares error (RMSE) of different correction methods.

**Figure 15 sensors-21-06283-f015:**
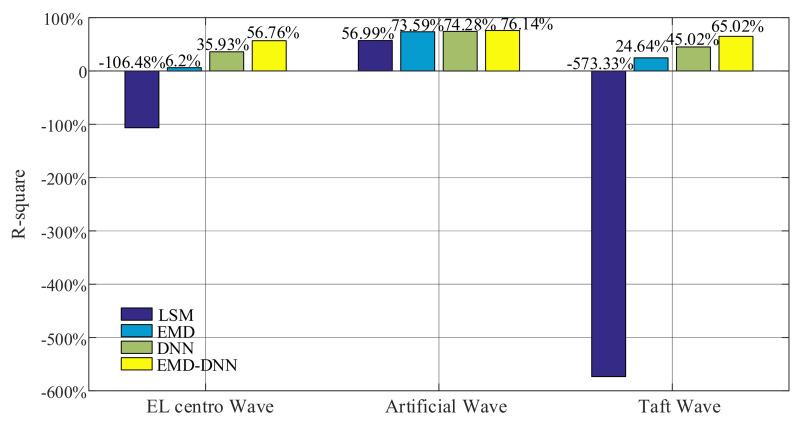
Fitting degree of different correction methods.

**Figure 16 sensors-21-06283-f016:**
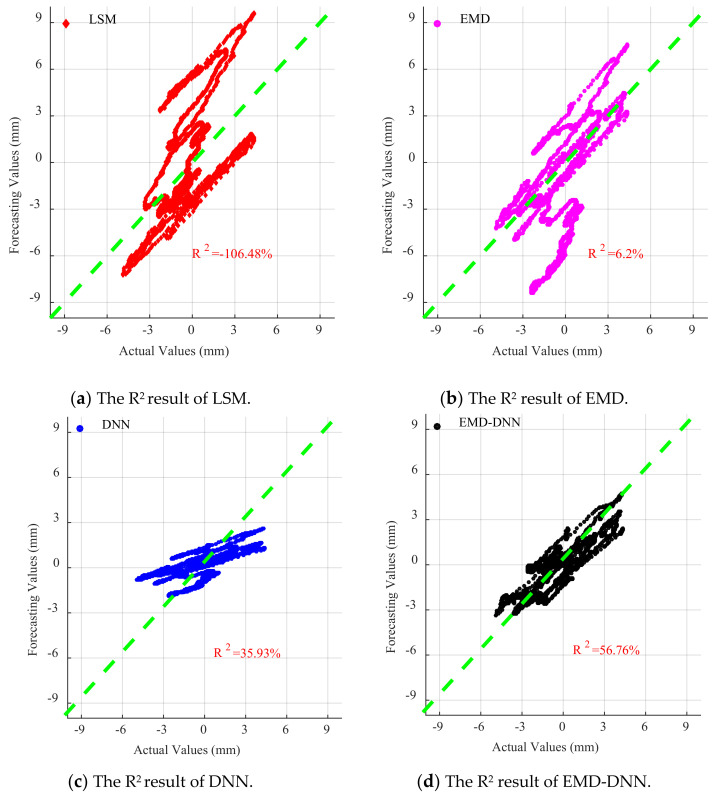
Scatter diagram of different correction methods for EL centro wave.

**Figure 17 sensors-21-06283-f017:**
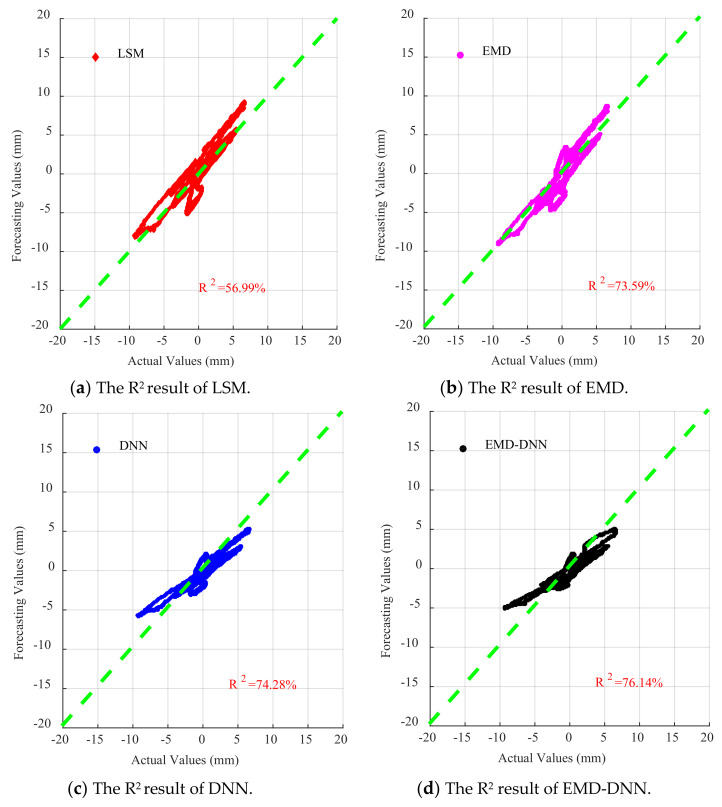
Scatter diagram of different correction methods for Artificial waves.

**Figure 18 sensors-21-06283-f018:**
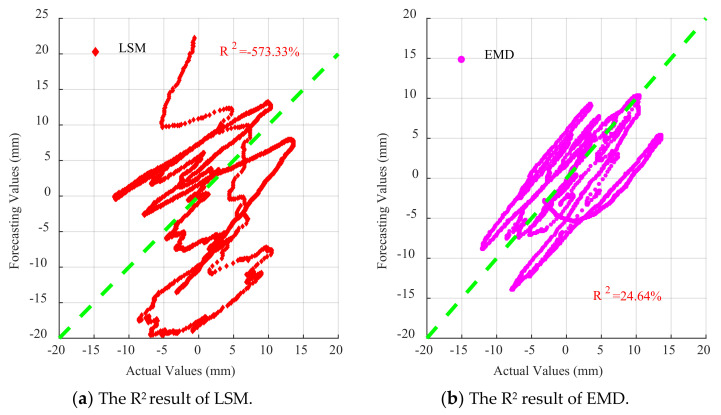
Scatter diagram of different correction methods for Taft wave.

**Table 1 sensors-21-06283-t001:** Summary of baseline correction methods.

Method	Time of Establishment	Advantages	Disadvantages
Iwan	1985	Simple and convenient	Parameter selection is fixed, only applicable to specific accelerometers
V0	2001	The selection of parameters has great physical meaning	Parameter calculation relies on subjective experience, which makes it easier to introduce errors
Wu and Wu	2007	The result of bilinear correction has high accuracy	Parameter selection depends on the preset threshold
Automatic iteration	2011	As far as possible to ensure the accuracy of parameter selection, reduce the subjective error	The calculation process is complicated, and the efficiency is relatively low

**Table 2 sensors-21-06283-t002:** Sensor layout of simulated seismic shaking table test.

Number	Sensor Type
A	ENDEVCO acceleration sensor
B	Unimeasure pull-wire displacement meter

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
