# Peer review of "Baseline Correction of Acceleration Data Based on a Hybrid EMD–DNN Method"

_sensors, 2021, doi:10.3390/s21186283_

Round 1

Reviewer 1 Report

see pdf

Author Response

Dear Editor,

Manuscript ID: sensors-1355735

Title: Baseline correction of Acceleration Data Based on EMD-DNN

Authors: Zengshun Chen, Jun Fu, Yanjian Peng, Tuanhai Chen, LiKai Zhang *, 
Chenfeng Yuan

Submitted to section: Sensor Networks

Please find attached a revised version of our manuscript, ‘Baseline correction of Acceleration Data Based on EMD-DNN’, which we resubmit for the consideration of publication in Sensors.

Thank you and the reviewer for the valuable comments, which enabled us to greatly improve the quality of our manuscript. According to the comments, we have made extensive modifications to the original manuscript. The modified texts were highlighted in the revised manuscript. Enclosed with this letter, in the following pages, are our point-by-point responses to each reviewer comment. We sincerely hope this manuscript will be finally accepted for publication. 

Thank you for coordinating the review processes of this manuscript. 

Yours truly,

L. K. Zhang, Doctor

School of Civil Engineering, 
Chongqing University
Shapingba District
Chongqing 400045, China
E-mail: zhanglikai@cqu.edu.cn

Reviewer 2 Report

Authors

This manuscript presents a useful approach to improve seismic signal analyses.

The abstract is well focused on the main problem the paper will be present and solve with the proposed approach. However, I would like to suggest to the authors to include a sentence to underline the limitation and difficulties of other approaches to baseline correction (as reported between lines 97 and 127 of the paper).

The main text is well structured and the methodology proposed is clear and well described, but more attention should be dedicated to equations and plots.

Specific comments:

Line 139 Should be (1)-(6)

Lines 140-1444 The sentences are not clear because the information about the earthquakes signals and shaking table configuration is provided later in the main text. Please rewrite the sentences to include more information.

Lines 207-209 Please specify the colour of recorded displacement

Figure 2 Please include the info about the red line.

Lines 239- 243 Please verify the variable into text and the equation. it doesn't fit

Figure 6 Should be improved. Please include a different view angle sketch reporting the information provided in the main text (lines 304-319)

Lines 346-357 Please include some sentences about Pro-Con of MAE; MSE; RMSE and R-square. Remove the repeated sentences

Figure 12 Please include (MAE) in the caption

Figures 13 and 14 Please include the full name of MSE in the caption

Figure 15 Please correct and describe the R2 label. Please justify why the values are expressed in per cent

Figures 16-18 Please check the uniformity of acronymous used in the main text and other figures

Overall, I think that this paper is well written and deserves publication with minor revision.

Author Response

(The authors gave the same response as above.)
